# Linking Context to Language Switching: Effects of Background Noise on Bilingual Language Comprehension

**DOI:** 10.3390/bs15010060

**Published:** 2025-01-13

**Authors:** Lu Jiao, Zejun Wang, Xiaoting Duan, Yingying Yu, Cong Liu

**Affiliations:** 1Department of Psychology, Qingdao University, Qingdao 266071, China; 2Brain, Cognition, and Language Learning Laboratory, Qingdao University, Qingdao 266071, China; 3Beijing Key Lab of Applied Experimental Psychology, Faculty of Psychology, Beijing Normal University, Beijing 100875, China; 4Center for Psychological Sciences, Zhejiang University, Hangzhou 310028, China

**Keywords:** bilingual, language switching, language comprehension, background noise

## Abstract

In the present study, we set out to examine the effect of background noise on bilingual language comprehension between a person’s first language (L1) and second language (L2). Language control systems seem to systematically engage in bilingual language production, as evidenced by the presence of switch costs with slower responses to switch trials than repeat trials. However, this systematic engagement does not uniformly apply to comprehension, as the involvement of language control in bilingual comprehension may vary depending on external contexts. In two experiments, we investigated how background noise influenced language switching in comprehension for unbalanced Chinese–English bilinguals. Overall, when comprehending words from two languages, participants experienced significant language switch costs across all conditions, but smaller switch costs were observed in the noise condition than in the quiet condition. However, the symmetrical patterns of switch costs were not modulated by background noise. This is the first study that supports the flexibility of bilingual language comprehension depending on the presence of background noise, expanding the adaptive control hypothesis.

## 1. Introduction

In the context of bilingual language control, the adaptive control hypothesis (ACH) posits that the control mechanisms employed by bilingual individuals are flexible. Specifically, the ACH distinguishes three types of language contexts (single-language, dual language, and dense-code switching) and emphasizes that the control system itself adjusts to different processing contexts, enabling bilinguals to communicate effectively in their intended language ([19]; [37]). Following this hypothesis, previous studies have found that language control is influenced by multiple processing contexts ([9]; [23], [24]). For example, [24] ([24]) combined a language-switching task with a semantic categorization task, and participants required bilinguals to decide if the meaning of a printed word belonged to a warm or cold color. In the non-conflicting condition, color words were presented in white ink (e.g., the word “yellow” in white), while the semantically related conflicting trials presented color words in an inconsistent color with the word’s meaning (e.g., the word “yellow” in green). The results revealed that the switch costs during bilingual language comprehension were larger in the semantic-conflicting context than in the non-conflicting context, supporting the influence of processing contexts on language switch costs. However, these studies were conducted in ideal, quiet conditions that could not be representative of natural communication contexts, such as homes and classrooms where bilinguals are exposed daily. Therefore, the present study examined whether bilingual language control in comprehension was influenced by the presence of background white noise.

The language-switching task has been widely used to investigate the language control mechanisms in the bilingual literature ([23]; [26]). There are two types of trials in such a task: the repeat trials, in which the intended languages of two subsequent trials are the same (e.g., L1-L1, L2-L2), and the switch trials, in which the intended languages of two subsequent trials are different (e.g., L1-L2, L2-L1). There are two common indicators of language control: one is *switch costs*, which was calculated by subtracting the latencies/accuracy of repeat trials from switch trials; another one is the *asymmetrical pattern of switch costs* compared between L1 and L2 ([10]; [24]). The psycholinguistic literature on bilingual language comprehension is always combined with the classical language-switching task with visually present words and a lexical recognition task ([28]), a picture-sentence matching task ([31]), a semantic decision task ([24]; [25]), or a number categorization task ([10]). For example, words from two languages are presented across trials and require bilingual participants to categorize each word (e.g., does the word refer to an animate or inanimate object?). It is typically found that comprehension-based switch trials yield a worse performance than repeat trials ([8]; [24]). For example, [28] ([28]) examined comprehension-based language control using a lexical decision task and revealed the presence of switch costs in comprehension and an asymmetrical pattern of costs. Moreover, [8] ([8]) implemented a practice block (in pure language) prior to a mixed switch block aiming to manipulate the relative activation of languages. The results of the language comprehension switch revealed that when L1 words were practiced, the switch costs and its asymmetrical pattern could be influenced. The presence of switch costs in bilingual comprehension tasks provided empirical evidence for the engagement of language control systems in bilingual language comprehension.

According to the Bilingual Interactive Activation model (BIA, see [17]; BIA-d, see [18]), language control in bilingual comprehension exhibits exogenous control driven by stimuli, which voluntarily triggers the target language node (e.g., language A) and inhibits non-target language interference (e.g., language B). Thus, when the switch trial presents a word of another language (e.g., language B) that has been inhibited in the previous trial, more time would be needed to release inhibition, resulting in the worse performance of the switch trial compared to the repeat trial. Moreover, the BIA+ model emphasizes that language control in bilingual comprehension occurs in decision systems where executive control processes are involved ([13]). These theoretical frameworks emphasize the importance of control processes during language switching in comprehension.

Studies on bilingual language comprehension have yielded mixed findings. For instance, behavioral research conducted a series of comprehension switching tasks by manipulating several variables (e.g., stimuli), but the switch costs did not always occur in bilingual language comprehension ([10]). Furthermore, combined with the adaptive control hypothesis ([19]), the processing context where bilinguals are communicating may potentially be associated with the language control mechanism in comprehension ([11]; [20]; [23]). One line of research linking context to language switching investigated whether semantically related contexts influence bilingual language control ([23], [24]). For example, one study revealed that semantic-conflicting contexts influenced the switch costs and the level of asymmetry. This study combined the bilingual language production with the color word Stroop. In the non-conflicting context, the non-color words were presented in red, blue, or green (e.g., “tap” word presented in red), while in the semantic-conflicting context, color words were presented in an inconsistent color (e.g., “green” word presented in red). The results revealed that the semantic noise induced by the color-word Stroop increased the switch costs compared to the non-conflicting context. Meanwhile, greater asymmetrical switch costs in the conflicting context were observed, showing larger switch costs in L1 than in L2 ([23]). Similarly, Liu and colleagues revealed the effect of semantic-conflicting contexts on switch costs from the bilingual language comprehension perspective ([24]).

The other line of research mainly linked background noise to language switching ([15]; [20]). [20] ([20]) required bilingual children to perform code-switching in a noise task. Children first heard single-language sentences and code-switched sentences and then answered a yes/no question to measure language comprehension. The quiet listening condition only presented sentences without noise, while, in the noise condition, speech-shaped noise was added to the sentence stimuli at a signal-to-noise ratio (SNR) of 0 dB (signal at 70 dB, noise at 70 dB). This study found comprehension costs between single-language and code-switched sentences in the noise condition. Moreover, one adult study employed Spanish–English bilinguals who were highly proficient in both languages to complete a bilingual word recognition task in a speech-shaped noise environment. The SNR was fixed at −14 dB (signal at 51 dB, noise at 65 dB). The findings showed that speech-shaped noise also reduced the word recognition performance in proficient bilinguals but with symmetrical switch costs between two languages ([15]). The poor performance of bilingual language switching in a noise environment may be related to the energetic masking phenomenon, which proposes the co-occurrence of signal and noise information hinders signal identification ([34]). Specifically, the presence of speech-shaped noise made the language switching more difficult, such that bilinguals have to spend more time to process masked speech recognition.

However, the stochastic resonance phenomenon in speech perception research proposes that additional noise information can augment the detection of weak or subthreshold speech information ([29]), such as improving sensory acuity in tone discrimination tasks ([5]), facilitating learning and memory ([33]). Some studies compared bilinguals and monolinguals in speech perception with background white noise, and bilinguals demonstrated earlier neural responses to speech sound compared to monolinguals in noise conditions ([3]). Therefore, to some extent, the neural encoding of speech in bilinguals may be more resistant to the detrimental effect of background noise ([3]; [29]; [33]). Additionally, it is worth noting that easier speech detection was always accompanied by white noise without speech-like information ([3]; [5]; [33]). Instead, the speech-shaped noise used in bilingual language control research was generated with a speech-like spectrum and speech-like fluctuation in amplitude ([15]; [20]). Thus, it is possible that bilingual language switching in comprehension can differ between quiet listening conditions and background white noise conditions. However, the available supportive evidence is far from conclusive.

In the present research, our goal is to examine whether background noise influences language control during bilingual language comprehension. We combined a traditional language-switching task with an animacy task, where participants identified if the words from two languages referred to something living or non-living ([10]). The unbalanced Chinese–English bilinguals with dominant Chinese were recruited to perform this task in a quiet condition and with background noise. In the quiet listening conditions, all words were auditorily presented with no noise, while in the noise condition, the target words were presented with white noise. The level of target words in both experiments was fixed at 70 dB. The level of white noise was 75 dB (SNR = −5 dB) in Experiment 1 but changed to 65 dB (SNR = 5 dB) in Experiment 2. According to the adaptive control hypothesis ([19]), we predicted that the language control pattern in a noisy environment would be different from previously observed patterns in quiet conditions. Furthermore, based on the energetic masking phenomenon, the language switching in noise would be more pronounced because the degradation of speech cues associated with masking reduces participants’ ability to process target words. By contrast, combined with the stochastic resonance phenomenon, the white noise, independent of linguistic/speech sound, augments signal detection and benefits bilingual language switching.

## 2. Experiment 1

### 2.1. Participants

The present study recruited thirty-six Chinese–English bilinguals. All participants were self-reported right-handed adults. They had normal or corrected-to-normal audio-vision ability. Before the experiment, all participants signed written informed consent and completed the language background questionnaire. The local ethics committee approved the study. We collected the age of L2 acquisition (AoA) and self-reported language proficiency in listening, speaking, reading, and writing skills on a 7-point scale (1 = very poor, 7 = excellent) (see Table 1). A paired-sample *t*-test on the final sample consisted of thirty-one participants and revealed that they were unbalanced bilinguals with dominant Chinese (L1: M = 6.40, SD = 0.80; L2: M = 3.65, SD = 0.58), *t* = 16.99, *p* < 0.001.

### 2.2. Materials

Participants performed an animacy judgment task where we required them to make a “living/nonliving” judgment for the presented words. We selected 60 words from the dataset of [35] ([35]). Thirty words referred to living entities, and thirty to non-living entities. One group of Chinese–English bilinguals were recruited to assess familiarity and animacy on a 5-point scale (1 = very unfamiliar/nonliving, 5 = very familiar/living). The assessment showed that they were familiar with all words (M = 4.67, SD = 0.16). And importantly, the animacy assessment showed a significant difference between living (M = 4.70, SD = 0.25) and nonliving conditions (M = 1.26, SD = 0.49), *t* = 34.17, *p* < 0.001. We asked one female Chinese native speaker to record both the Chinese and the English words. Using Praat, the recordings of Chinese and English words were fixed at 70 dB. The intensity of white noise was fixed at 75 dB without any linguistic information. Another group of participants was required to assess whether the background noise masked signal words on a 4-point scale, and this SNR (−5 dB) showed a significant masking effect compared to quiet listening conditions (*t* = 7.62, *p* < 0.001).

### 2.3. Task and Procedure

Experiment 1 was designed as a 2 (condition: noise vs. quiet) × 2 (language: L1 vs. L2) × 2 (type: repeat vs. switch) within-subject experiment. Firstly, participants were required to familiarize themselves with the L1 and L2 picture names in order to reduce error rates. Then, they were required to complete the language-switching task with an animacy judgment task. The experiment consisted of a quiet block (70 dB signal with no noise) and a noise block (70 dB signal accompanied by 75 dB noise), and the presentation order was counterbalanced across participants. There was a total of 121 trials in each block, where the first trial was a filler trial. Therefore, there are 30 trials for each trial type (i.e., L1-L1 repeat, L2-L2 repeat, L2-L1 switch, and L1-L2 switch). In each listening condition, all trials were pseudorandomized with no more than four trials of the same type (switch or repeat), language (L1 or L2), or animacy category in a row.

Each trial began with a fixation cross presented in the center of the screen for 500 ms, followed by an aurally presented target word in Chinese or English. Participants were instructed to make a “living” judgment by pressing the key “F” if the word referred to something living (e.g., animal or plant) or part of a living thing (e.g., ear) and to make a “non-living” judgment by pressing the key “J” if the word referred to something non-living (e.g., chair). The response keys were counterbalanced across participants. Once a response was given or after a maximum duration of 2000 ms, a blank screen was presented for 500 ms prior to the next trial. Before the formal experiment, there was a practice block consisting of 8 trials.

### 2.4. Data Analyses

Five participants in Experiment 1 were excluded from the final data analysis because their accuracy was lower than 80%. Thus, the final sample consisted of thirty-one participants (18 females) with an average age of 19.52 years old (SD = 1.43). The response times (RTs) in the language-switching task were analyzed using frequentist analyses and Bayesian regression analyses. RTs were log-transformed as dependent variables. For frequentist analyses, the linear mixed-effects models were conducted in R using the *lme4* package ([2]). The fixed effects included condition (quiet, noise), language (L1, L2), type (repeat, switch), and their full interactions with subjects and items as random effects. All variables were sum-contrast-coded (i.e., −0.5 for quiet and 0.5 for noise; −0.5 for L1 and 0.5 for L2; −0.5 for repeat and 0.5 for switch).

The Bayesian regression analyses with multilevel Bayesian regression modeling were also conducted on response times in R using the *brms* package ([4]). For Bayesian regression analyses, our modeling contained the same fixed effects as the frequentist analyses but used a maximal random effects structure, which often fails to converge in frequentist analyses. A key advantage of Bayesian regression modeling over frequentist analyses is that it presents the whole posterior probability distribution of each effect rather than only a binary test for the null hypothesis. There are two common indicators in Bayesian models: *credibility intervals* and *evidence ratios* in favor of a directional hypothesis. Evidence ratios of 19 are interpreted as significant at α = 0.05 for a given hypothesis ([27]), with those larger than 30 representing “very strong” evidence ([14]). Meanwhile, if the 95% credibility interval does not contain zero (i.e., less than 2.5% of the posterior distribution located on the other side of zero), a given hypothesis could also be viewed as “significant” at an α = 0.05 level ([27]). Moreover, Bayesian regression modeling also incorporates prior knowledge. Largely in line with previous relevant studies, we used weakly informative priors that constrained the models to psycholinguistically plausible parameter estimates ([16]; [22]). The priors for fixed effects followed a normal distribution of N (0, 1). We derived the posterior distributions based on the Hamiltonian Monte Carlo with four Markov chains of 10,000 iterations each, including 1000 warm-up iterations. All parameters had a Gelman–Rubin statistic Rhat equal to 1.00. Prior to analyses, we excluded trials with error responses, trials faster than 200 ms, and RTs more than 2.5 standard deviations (SDs) from the mean ([7]; [21]).

### 2.5. Results

#### 2.5.1. Frequentist Analyses on Experiment 1

Figure 1 and Table 2 present RTs of the language switching task in the quiet/noise conditions of Experiment 1. The fixed effects of the RTs model included condition, language, type, and interactions; the random effects included the by-subject random slope for condition and language and the by-item random slope for condition. Table 3 presents the fixed effects structure for the linear mixed-effect model in Experiment 1. The main effect of this condition was significant, with slower responses in the noise condition than in the quiet condition; the significant main effect of language showed slower responses in L2 words compared to L1 words; and the significant main effect of the types showed that the RTs in switch trials were slower than in repeat trials.

In addition, the two-way interaction of condition × language was significant, showing a smaller difference between the two languages for the noise condition than in the quiet condition. The significant interaction of condition × type showed a smaller switch cost (i.e., the difference between switch and repeat trials) for the noise condition than in the quiet condition. Despite the fact that the two-way interaction between language and type was not significant, the three-way interaction reached significant levels. Further analysis revealed that the switch costs between L1 and L2 were a symmetrical pattern in the noise condition (Estimated = −0.05, SE = 0.10, *t* = −0.53, *p* = 0.59), whereas in the quiet condition, the switch costs were an asymmetrical pattern of L1 and L2 (Estimated = 0.22, SE = 0.09, *t* = 2.40, *p* = 0.02).

#### 2.5.2. Bayesian Analyses on Experiment 1

Table 4 summarizes the results of the Bayesian analyses on the RTs of Experiment 1. In line with the findings from the frequentist analyses, we observed the significant main effects of the condition (*ER* = 366.35, *PP* = 1.00), language (*ER* > 9999, *PP* = 1.00), and type (*ER* = 30.80, *PP* = 0.97). Meanwhile, the two-way interactions of condition × language (ER = 2768.23, PP = 1.00) and of condition × type (*ER* = 44.63, *PP* = 0.98) both reached a significant level. Interestingly, the three-way interaction (condition × language × type) was not significant (*ER* = 8.62, *PP* = 0.90), with the 95% CI containing zero (see Appendix A).

### 2.6. Discussion of Experiment 1

Experiment 1 first showed the main effect of type, suggesting that the unbalanced Chinese–English bilinguals experienced switch costs during bilingual language comprehension. The presence of switch costs was consistent with previous research revealing switch costs in auditory comprehension tasks ([7]; [24]), as opposed to some findings of no switch costs ([10]). More importantly, we observed that the background noise affected the switch costs in bilingual language comprehension, as evidenced by the significant interaction between condition and type. Specifically, the switch costs with the presence of white noise were smaller than those in a quiet condition, which can be explained by the stochastic resonance phenomenon. Based on the stochastic resonance phenomenon, additional information, such as the white noise in the present study, could augment the detection of a weak speech stream ([29]). Therefore, the presence of white noise might increase sensory acuity and benefit signal detection. Consequently, participants could be more sensitive to perceiving and recognizing the signal information (i.e., target words).

Previous research on speech perception in noise always included two or more levels of signal-to-noise ratios (SNRs). For example, [38] ([38]) required adults to identify single words in a quiet condition, a moderately loud noise condition (SNR at 20 dB), and a loud noise condition (SNR at −5 dB). Moreover, [32] ([32]) asked participants to hear single-language sentences and code-switching sentences with the presence of white noise at four SNR conditions, including −6, −3, 0, and 3 dB. Based on these studies, we further investigated bilingual language control in comprehension with a different SNR condition, where the signal intensity was fixed at 70 dB, which is the same as Experiment 1, but the noise intensity changed to 65 dB.

## 3. Experiment 2

### 3.1. Participants

Another group of 35 Chinese–English bilinguals participated in Experiment 2. All participants had a normal or corrected-to-normal audio–vision ability. The same as Experiment 1, we collected participants’ language background in L2 AoA and language proficiency (see Table 5). A paired-sample *t*-test between L1 and L2 revealed that participants of Experiment 2 were also unbalanced bilinguals with dominant Chinese (L1: M = 6.33, SD = 0.75; L2: M = 3.88, SD = 0.67), *t* = 13.44, *p* < 0.001. Moreover, we conducted the independent-sample *t*-tests for each language skill between Experiment 1 and 2, and the *t* values for the comparison of participants’ language background between Experiment 1 and Experiment 2 were not significantly different (*p* > 0.05).

### 3.2. Material and Task

The experimental materials and procedures of Experiment 2 were the same as those in Experiment 1, but the noise condition was changed to a signal intensity of 70 dB with a white noise background of 65 dB. The assessment of whether the background noise masked signal words also revealed a significant masking effect (*t* = 4.13, *p* = 0.001).

### 3.3. Data Analyses

Six participants were excluded because their accuracy was lower than 80%, leaving twenty-nine participants entering the final analyses (13 females) with an average age of 20.10 years old (SD = 0.99). The RTs in Experiment 2 were analyzed using frequentist analyses and Bayesian regression analyses, which are the same as that of Experiment 1.

### 3.4. Results

#### 3.4.1. Frequentist Analyses of Experiment 2

The RTs of the language switching task in the quiet and noisy conditions of Experiment 2 are presented in Figure 2 and Table 6. The fixed effects of the RTs model included condition, language, type, and interactions; the random effects included the by-subject random slope for condition and language and the by-item random slope for condition and type. Table 7 summarizes the fixed effects structure for the linear mixed-effect model in Experiment 2. Despite the fact that there was no main effect of the condition, the main effects of language and type were both significant. Moreover, the interaction of condition × type was significant, showing a smaller switch cost in the noise condition than in the quiet condition. The significant interaction of condition × language showed that differences in RTs between the two languages were smaller in the noise condition than in the quiet condition. The three-way interaction was not significant, suggesting that the switch costs between L1 and L2 were symmetrical in quiet and noisy conditions.

#### 3.4.2. Bayesian Analyses of Experiment 2

Table 8 summarizes the results of the Bayesian analyses on the RTs of Experiment 2. In line with the findings from the frequentist analyses, the main effects of language (*ER* > 9999, *PP* = 1.00) and type (*ER* = 492.15, *PP* = 1.00) were significant but absent in the main effect of the condition (*ER* = 0.74, *PP* = 0.43). Moreover, the two-way interactions of condition × language (*ER* = 115.88, *PP* = 0.99) and of condition × type (*ER* = 100.98, *PP* = 0.99) both reached significant levels. The two-way interaction (language × type, *ER* = 0.19, *PP* = 0.16) and the three-way interaction (condition × language × type, *ER* = 1.44, *PP* = 0.59) were not significant, with 95% CIs containing zero (see Appendix B).

### 3.5. Discussion of Experiment 2

Experiment 2 showed the significant effect of type, in line with Experiment 1, indicating the occurrence of switch costs during bilingual language comprehension tasks ([7]; [30]). Moreover, the significant interaction between condition and type provided further evidence for the effect of background noise, showing a reduced switch cost in the noise condition relative to the quiet condition. However, inconsistent with Experiment 1, the main effect of the condition was not significant in Experiment 2. Combined with the energetic masking phenomenon ([34]), the weaker noise compared to signal information might not hinder target word identification, showing parallel response speeds across noisy and quiet conditions.

## 4. General Discussion

Across two experiments, this study examined whether and how background noise influenced language switching in comprehension. To achieve this, unbalanced Chinese–English bilinguals performed an animacy judgment task where words from two languages were presented across trials. The SNR was different between the two experiments. It was fixed at −5 dB (75 dB for noise and 70 dB for signal) in Experiment 1, whereas it was fixed at 5 dB in Experiment 2 (65 dB for noise and 70 dB for signal). Overall, we found that Chinese–English bilinguals experienced significant switch costs across quiet listening conditions and background noise conditions. Moreover, there was strong evidence for the background noise effect on bilingual language comprehension, showing that background noise reduced the switch costs in comprehension relative to the quiet condition through two experiments. However, there was no consistent evidence for the background noise effect on the asymmetrical pattern between L1 and L2 switch costs.

The presence of switch cost in comprehension was in line with the models of bilingual word recognition postulating non-selective lexical access in bilingual language comprehension (BIA+, see [13]; BIA-d, see [18]). During bilingual language comprehension, the lexical candidates from two languages were both activated and competed for selection and recognition ([36]). Specifically, the activation of a particular language node allowed for the selection of words in that language while inhibiting words in other languages. As specified in the Bilingual Interactive Activation (BIA) model, there is a top-down control from language nodes within the language system in bilingual word comprehension ([12]). Thus, when it comes to switching trials (e.g., from L1 to L2), the recognition of the current target language (L2), which was inhibited in previous trials, could consume more cognitive resources to perform correctly compared to repeat trials, resulting in the occurrence of switch costs. The switch costs during bilingual language comprehension also appeared in previous empirical studies ([7]; [30]). For example, in the first experiment of the Aparicio and Lavaur study ([1]), French–English bilinguals responded slower to switch trials than repeat trials in a lexical decision task.

More importantly, our study revealed that background white noise influenced switch control during bilingual language comprehension. Two experiments with different noise intensities consistently showed that, compared to a quiet listening environment, the switch costs in language comprehension switching tasks were reduced in the white noise condition. The adaptive control hypothesis distinguishes three language contexts and emphasizes that the language control system in bilinguals can adapt to these varied language contexts ([19]). Incorporating non-linguistic information (i.e., the white noise) into the adaptive control hypothesis, our findings extend the adaptive control hypothesis into more naturalistic settings where bilinguals are exposed daily. In the present study, the quiet and noise conditions both involved two languages, and participants had to resolve the interference between these two languages in order to avoid cross-language intrusion errors. Importantly, the presence of white noise increased the interference control demand, thus triggering the engagement of additional cognitive processes for bilingual language comprehension.

However, the patterns of results in the white noise condition were inconsistent with some previous studies ([24]). For example, despite [24] ([24]) also found the adaptive changes in language control mechanisms, the effect of conflicting information was opposite to our findings, showing a higher switch cost in the conflicting condition. The discrepancy between our study and [24] ([24]) may be associated with the type of noise information. [24] ([24]) manipulated conflicting conditions through the semantic conflicts between the meaning and ink color of target color words, providing linguistically related noise information on bilingual language comprehension. By contrast, the background white noise as conflicting information in the present study was not involved in semantic processing. Furthermore, from the perceptual processing perspective, the presence of white noise might benefit sensory acuity and improve signal detection ([5]) (see stochastic resonance phenomenon ([29]). For switch trials (e.g., from L1 to L2), individuals in noise conditions could be more sensitive to recognizing the target language (e.g., L2), which is at a relatively lower activation level (due to the L1 overactivation in the previous trial). Therefore, there is less time for the target language to reach the activation threshold and make animacy judgments, reducing the switch costs in noise conditions. Therefore, such variations might lead to discrepancy effects of conflicting information on the switch costs in comprehension.

The other core question in the present study focused on the asymmetrical patterns of comprehension-based switch costs in quiet versus noise conditions. Experiment 1 showed a numerically higher switch cost in L2 than in L1; however, the more rigorous analysis by Bayesian regression revealed a symmetrical pattern between L1 and L2, as evidenced by the absent three-way interaction between condition, language, and type. This symmetry can be driven by the improved sensory acuity, facilitating signal detection and the engaged cognitive control benefiting inhibition release ([5]; [29]; [33]). Furthermore, we observed the main effect of the condition and the interaction between the condition and language. To some extent, these findings were in line with our hypothesis because the presence of noise could interfere with the overall response speed. Given the unbalanced bilinguals employed in the present study, the less dominant L2 possibly further hindered speed judgment, resulting in slower response times in L2 compared to L1 ([6]). However, noise effects on overall comprehension performance were only observed in Experiment 1, with the significant main effect of the condition variable, which was not found in Experiment 2. Garcia and colleagues (2018) also revealed that speech-shaped noise reduced word recognition performance in bilinguals. In line with Experiment 1 in the present study, the SNR in Garcia’s study was fixed at −14 dB (signal at 51 dB, noise at 65 dB), where the noise intensity was larger than the signal intensity. Combined with the masking phenomenon ([34]), the presence of noise information may hinder signal perception and identification. On the other hand, in Experiment 2, the weaker noise intensity may be inadequate to mask target signals, resulting in a similar performance in overall response speed between quiet and noise conditions.

One limitation of the present study is that two SNR settings were operated between two groups of participants. Despite there being no significant differences between participants in their language background, it would be better to examine the SNR effects using a within-participant design in future studies. Moreover, the present study manipulated the SNR by increasing or decreasing 5 dB relative to a single intensity. Given the masking phenomenon, it is unclear whether the greater SNR could trigger the interfering effect of white noise on bilingual language control.

## 5. Conclusions

In sum, our data showed the effect of background noise on switch costs in comprehension, but an absent effect on the asymmetrical pattern between L1 and L2 costs was observed. This suggests that, at least for some bilinguals, language control does engage in bilingual language comprehension, supported by the stable presence of switch costs. Importantly, the switch costs during bilingual language comprehension could be affected by the presence of white noise. For all we know, the present study is the first to observe the switch costs in comprehension with a white noise background where bilinguals are often communicating. These findings support and expand the adaptive control hypothesis ([19]).

## Figures and Tables

**Figure 1 behavsci-15-00060-f001:**
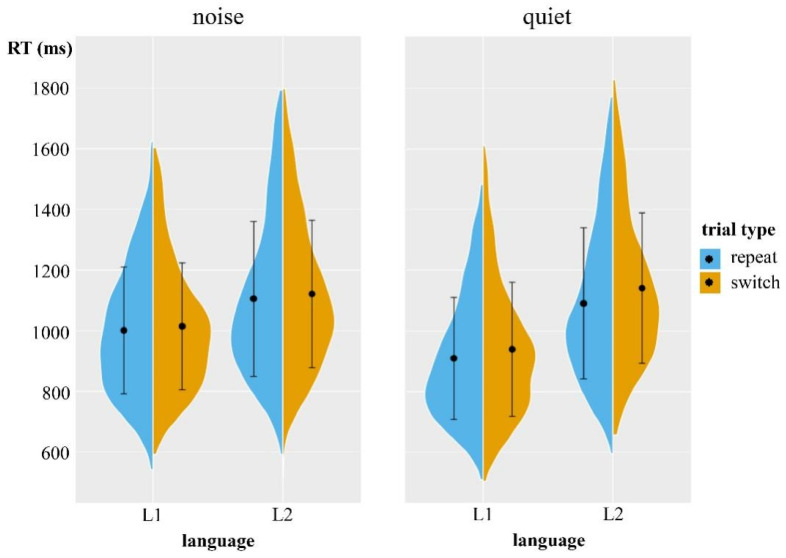
Split violin plots showing the RTs of the language switching task in Experiment 1. The black dots represent the mean value, and the vertical black lines represent the standard deviation.

**Figure 2 behavsci-15-00060-f002:**
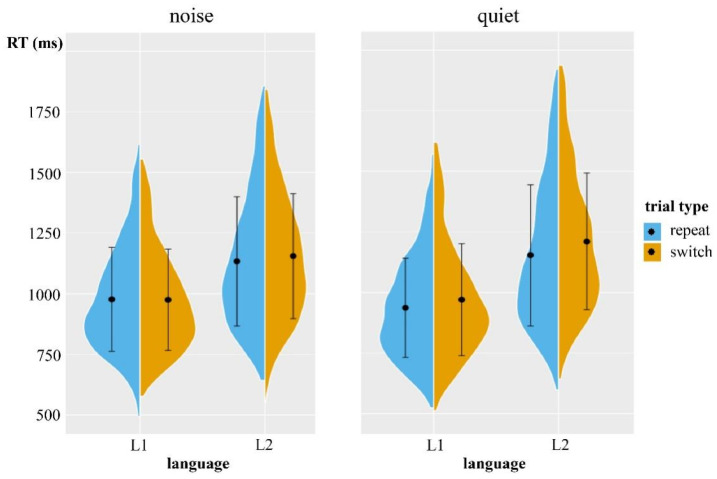
Split violin plots showing the RTs of the language switching task in Experiment 2. The black dots represent the mean value, and the vertical black lines represent the standard deviation.

**Table 1 behavsci-15-00060-t001:** Means (SDs) of AoA and language proficiency in Experiment 1.

	L1	L2
AoA (years)	-	8.23 (3.18)
Self-reported proficiency (1–7)		
Listening	6.48 (0.89)	3.55 (0.99)
Speaking	6.45 (0.92)	3.35 (0.66)
Reading	6.45 (0.81)	4.13 (0.92)
Writing	6.23 (0.88)	3.58 (0.76)

**Table 2 behavsci-15-00060-t002:** RTs and switch costs (SDs) in Experiment 1.

	L1			L2		
	Repeat	Switch	Switch Costs	Repeat	Switch	Switch Costs
quiet	910.94 (87.54)	941.46 (92.20)	30.53 (44.63)	1092.98 (91.96)	1144.65 (108.59)	51.68 (47.72)
noise	1004.98 (100.86)	1017.53 (103.70)	12.55 (49.64)	1109.96 (122.54)	1123.87 (122.89)	13.91 (41.95)

**Table 3 behavsci-15-00060-t003:** Estimates of fixed effects for the linear mixed-effect model in Experiment 1.

Fixed Effects	Estimated	SE	*t*	*p*
(intercept)	0.06	0.07	0.93	0.35
condition	0.19	0.07	2.75	<0.01
language	0.63	0.08	7.85	<0.01
type	0.16	0.03	4.93	<0.01
condition × language	−0.34	0.10	−3.52	<0.01
condition × type	−0.23	0.06	−3.48	<0.01
language × type	0.08	0.06	1.23	0.22
condition × language × type	−0.28	0.13	−2.17	0.03

**Table 4 behavsci-15-00060-t004:** Summary of the directional hypothesis for the effects in Experiment 1.

Family: Gaussian Links: mu = identity Formula: RT ~ condition × type × language + (condition × type × language|participant) + (condition × type × language|item) Data: Experiment 1 data Samples: 4 chains, each with iter = 10,000; warmup = 1000; thin = 1; total post-warmup samples = 36,000							
**Hypothesis**	**Estimated**	**SE**	**95% CI**		**Evid.** **Ratio**	**Post.** **Prob**	**Star**
condition > 0	0.20	0.07	0.08	0.32	366.35	1.00	*
language > 0	0.63	0.09	0.49	0.77	>9999	1.00	*
type > 0	0.10	0.05	0.01	0.18	30.80	0.97	*
condition × language < 0	−0.33	0.10	−0.49	−0.17	2768.23	1.00	*
condition × type < 0	−0.18	0.09	−0.33	−0.03	44.63	0.98	*
language × type < 0	0.04	0.10	−0.12	0.20	0.56	0.36	
condition × language × type < 0	−0.23	0.18	−0.52	0.07	8.62	0.90	

Note: Columns from left to right: the hypothesis being tested; the estimated mean of the standardized effect; the standard error; the 95% credibility interval; the evidence ratio in favor of the hypothesis; the posterior probability for the hypothesis; and a star, if 0 lies outside the 95% CI.

**Table 5 behavsci-15-00060-t005:** The means (SDs) of AoA and language proficiency in Experiment 2.

	Experiment 2		Independent-Samples *t*-Test	
	L1	L2	L1	L2
AoA(years)	-	*7.76 (1.88)*	-	0.69
Self-reported proficiency (1–7)				
Listening	6.48 (0.73)	*3.96 (1.08)*	0.05	−1.55
Speaking	6.31 (0.85)	*3.55 (0.83)*	0.61	−1.50
Reading	6.45 (0.78)	*4.28 (1.03)*	0.02	−0.58
Writing	6.07 (0.99)	*3.72 (0.96)*	0.65	−0.64

Note: independent-sample *t*-test = the *t* values for the comparison of participants’ language background between Experiment 1 and Experiment 2.

**Table 6 behavsci-15-00060-t006:** RTs and switch costs (SDs) in Experiment 2.

	L1			L2		
	Repeat	Switch	Switch Costs	Repeat	Switch	Switch Costs
quiet	939.14 (91.80)	972.80 (101.40)	33.67 (45.03)	1159.66 (117.21)	1216.62 (131.74)	56.96 (61.94)
noise	979.75 (102.31)	977.48 (101.41)	−2.28 (43.77)	1136.38 (135.25)	1160.99 (126.28)	24.62 (47.91)

**Table 7 behavsci-15-00060-t007:** Estimates of fixed effects for linear mixed-effect model in Experiment 2.

Fixed Effects	Estimated	SE	*t*	*p*
(intercept)	0.06	0.08	0.80	0.43
condition	−0.01	0.06	−0.05	0.96
language	0.79	0.09	8.92	<0.01
type	0.15	0.04	3.40	<0.01
condition × language	−0.22	0.07	−3.13	<0.01
condition × type	−0.21	0.07	−3.05	<0.01
language × type	0.10	0.09	1.18	0.24
condition × language × type	−0.02	0.14	−0.13	0.90

**Table 8 behavsci-15-00060-t008:** Summary of the directional hypothesis for the effects in Experiment 2.

Family: Gaussian Links: mu = identity Formula: RT ~ condition × type × language + (condition × type × language|participant) + (condition × type × language|item) Data: Experiment 2 data Samples: 4 chains, each with iter = 10,000; warmup = 1000; thin = 1; total post-warmup samples = 36,000							
**Hypothesis**	**Estimated**	**SE**	**95% CI**		**Evid.** **Ratio**	**Post.** **Prob**	**Star**
condition > 0	−0.01	0.07	−0.12	0.10	0.74	0.43	
language > 0	0.78	0.09	0.63	0.94	>9999	1.00	*
type > 0	0.13	0.04	0.06	0.21	492.15	1.00	*
condition × language < 0	−0.19	0.08	−0.33	−0.06	115.88	0.99	*
condition × type < 0	−0.20	0.08	−0.34	−0.06	100.98	0.99	*
language × type < 0	0.09	0.09	−0.06	0.23	0.19	0.16	
condition × language × type < 0	−0.04	0.17	−0.31	0.24	1.44	0.59	

Note: Columns from left to right: the hypothesis being tested; the estimated mean of the standardized effect; the standard error; the 95% credibility interval; the evidence ratio in favor of the hypothesis; the posterior probability for the hypothesis; and a star, if 0 lies outside the 95% CI.

## Data Availability

The data in this study are available from the corresponding authors upon reasonable request.

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
