# Peer review of "Linking Context to Language Switching: Effects of Background Noise on Bilingual Language Comprehension"

_behavsci, 2025, doi:10.3390/bs15010060_

Round 1

Reviewer 1 Report

Comments and Suggestions for Authors

The reviewer's comments in the PDF File.

Author Response

Response to Reviewer 1

This study examined the effect of context on bilingual language switching and its switching cost in general conflict contexts, and the results showed that there were similar symmetrical effects of language switching costs across two experiments. This study is helpful for expanding the adaptive control hypothesis of language switching based on language understanding in different sound environments. However, there are still some problems that need to be carefully revised by the authors.

Introduction

1.Lacking of literature support for the selection of key variables and their relationships in the Introduction.

Firstly, the author said, “This manipulation is based on the associations between white noise and perceptual processing/cognitive control processing, which are closely related with bilingual language control”, what is the specific link between white noise and bilingual control? How is the relationship between these factors in previous literature? And in paragraph 1, “Following this hypothesis, previous studies have found that language control was influenced by multiple processing contexts”. The authors mentioned that there had been theoretical and empirical studies proving that context had an effect on bilingual language control, but why did author after said that “However, there is little evidence for such effects considering the white noise context that bilinguals encounter in daily life.” And this statement had not supported by relative literatures.

Secondly, the last paragraph of the Introduction presented that “we expected the different amounts of conflicts may exert various effects”, however, the Introduction section did not provide a detailed literature analysis of whether white noise volume levels were representative of conflict contexts and the effects of volume levels on language control.

  • We thank the reviewer for these valuable comments. Firstly, in the revised Introduction section, we have introduced two lines of research linking context to bilingual language control, including the semantically-related context effect and the speech-shaped noise effect (Lines 84-114). Different from the speech-shaped noise with speechlike spectrum and speechlike fluctuation in amplitude, white noise is independent of speech and linguistic information. It is still unclear the potential role of background white noise in bilingual language control in comprehension, a question tested in the present study. Second, we have removed the phrasing of "...amount of conflicts", and revised the hypothesis.

2.The Introduction section is illogical and simply lists the literature and does not provide a focused literature analysis of the problem.

For example, In Para.3,Almost all of the literature listed speak “The language-switching task”, but the end abruptly suggested that “Overall, several empirical studies and theoretical frameworks emphasize the application of language control systems in bilingual language comprehension”.

  • We thank the reviewer for the constructive comment. We have revised the Instruction section in the revised manuscript (Lines 61-67).
  1. Important concepts are not explained in Introduction.

For example, what is “domain-general conflicting contexts”? And what is “amounts of conflicts”? Could amounts represent the white noise volume?

  • We thank the reviewer for the valuable comments. Combined with previous studies, we have changed the phrasing of "domain-general conflicting context" into "background noise/the presence of noise", and have removed the phrasing of "...amounts of conflicts" in order to avoid misunderstanding. Moreover, we examined the different background noises by operating signal-to-noise ratios (SNR). The level of signal (i.e., target words) in both experiments was fixed at 70 dB. The level of white noise was 75 dB (SNR = -5 dB) in Experiment 1 but changed to 65 dB (SNR = 5 dB) in Experiment 2.

Experiment 1

1.Participants. Five participants were excluded from the final data analysis because their accuracy was lower than 80%.” The elimination of subjects based on accuracy should be written in the Result or Data Analysis section, and Participants section should only write about how subjects were selected and the criteria for selection.

  • Thanks for the valuable comment. In the revised manuscript, we have rephrased "Participants" section (Lines 149-157), and added the elimination criteria of participants in the "Data Analysis" section.
  1. Task and Procedure. The author could add the experiment plot to let procedure become clearer.
  • We agree with the reviewer that the procedure plot could be helpful for the readers, but the aurally presented target words and background noise were hard to bring out in a plot. We have clarified the experimental procedure in the revised manuscript (Lines 185-192).
  1. Material. Experiment 1 did not perform material evaluation of vocabulary and white noise. We hope the author could be able to add it.
  • Thanks for the valuable comment. In the revised manuscript, we have added more details on the materials evaluations (Line 158-172).

4.Results. What data is being analyzed by Bayesian analysis is not specified in the section. As well as Experiment 1 filtered subjects based on accuracy but did not analyze the accuracy data, may I ask why did the author not use the three standard deviation screening RT for subject data filtering?

  • First, we analyzed the RTs data with Bayesian analysis (Line 205). Based on previous studies (e.g., Coumel, Liu, Trenkic, & de Bruin, 2024), we did not analyze accuracy as these data mostly represented wrong animacy category judgment. Second, the data filtering criterion (2.5 standard deviations in the present study) was based on relevant studies, e.g., the bilingual language comprehension in Coumel et al. (2024), and the bilingual language production in Jiang et al. (2024).

5.Discussion. Author's presumed cause does not correspond to the cited literature. The author thought that the context effects of white noise observed in Experiment 1 was caused by different amount of conflicts. However, citing Liu’s paper does not explain the amount of conflicts could influence it but Liu just differs from the author only in types of stimuli, there is no clearly identified stimuli quantitative difference. Therefore, what is the basis for distinguishing the volume (or amount) of noise in Experiment 2?

  • Thanks for the valuable comment. In the revised manuscript, we have revised this section (Line 276-284).

Experiment 2

1.Procedure. Experiment 2 sets the 70dB and 65dB to represent the different amounts of conflicts between two groups, but is there a significant difference between 70dB and 65dB in volume level? Did the white noise material found other people to evaluate?

  • Thanks for the valuable comment. We have evaluated the background noise materials and found a significant difference between 70 dB and 65 dB (Line 302).

2.Discussion. For “additional analysis by comparing switch costs”, the author did not discuss results and found previous studies to analyze it.

  • As suggested by reviewer 3, we have removed the "exploratory analysis" section and also removed these related discussions in the revised manuscript.

General Discussion

  1. The General Discussion did not analyze the reasons for important results. Why the author did not discuss the result just like “However, inconsistent with Experiment 1 the main effect of context was not significant in Experiment 2.”?
  • We thank the reviewer for the valuable comment. In the revised manuscript, we have added a detailed discussion of this result (Lines 418-427).
  1. The results in Experiment 2 conflicted with the General Discussion’s write. In Experiment 2 Discussion section, the author said there was no main effect of context, but why could say “our findings suggest that the conflicting contexts where bilinguals are communicating could adaptively modulate the language control in comprehension” in General Discussion?
  • Thanks for the valuable comment. Given the bilingual language control was measured by switch costs (i.e., the differences between switch and repeat trials), the interaction between condition and type variables reflected the background noise effects on language control, with significant such interactions in both two experiments. Whereas, the main effect of the condition reflected the background noise effects on overall performance in word comprehension, which was only significant in Experiment 1. In order to avoid misunderstanding, we have removed this sentence and discussed the results in the revised manuscript.

Once again, we thank the reviewer very much for spending the time reviewing our work.

Reviewer 2 Report

Comments and Suggestions for Authors

This study conducted two experiments to examine the role of domain-general conflicting contexts in comprehension-based language switching. The results interestingly reveal the presence of switch costs across all contexts, and importantly a reduced switch cost in conflicting noise context than non-conflicting quiet context. This study enriches our understanding on the flexibility of comprehension-based language control mechanism. However, there are still further issues that need to be addressed:

1. In introduction, the study of Declerck & Grainger (2017) was cited, but the description was a bit vague, so please describe it in detail.

2. Could the author provide a more-detailed hypothesis on the presence of comprehension-based switch cost and its asymmetrical pattern?

3. It would help readers better understand if the authors provided more details on the experimental materials.

4. One contribution of this study is to examine the flexibility of comprehension-based language control by operating the conflicting amounts. I believe it would be better to elaborate the rationale behind two experiments and inconsistent findings between them.

5. In discussion, the authors should further discuss the findings in combination with the adaptive control hypothesis.

Comments on the Quality of English Language

It will be better if language can be improved. 

Author Response

Response to Reviewer 2

This study conducted two experiments to examine the role of domain-general conflicting contexts in comprehension-based language switching. The results interestingly reveal the presence of switch costs across all contexts, and importantly a reduced switch cost in conflicting noise context than non-conflicting quiet context. This study enriches our understanding on the flexibility of comprehension-based language control mechanism. However, there are still further issues that need to be addressed:

  1. In introduction, the study of Declerck & Grainger (2017) was cited, but the description was a bit vague, so please describe it in detail.
  • Thanks for the valuable comment. We have elaborated on the study of Declerck & Grainger (2017) in the revised manuscript (Lines 61-67).
  1. Could the author provide a more-detailed hypothesis on the presence of comprehension-based switch cost and its asymmetrical pattern?
  • We thank the reviewer for the valuable suggestion. In the revised manuscript, we have added a more detailed hypothesis (Lines 138-145).
  1. It would help readers better understand if the authors provided more details on the experimental materials.
  • We have provided more details on the experimental materials evaluation (Lines 158-172).
  1. One contribution of this study is to examine the flexibility of comprehension-based language control by operating the conflicting amounts. I believe it would be better to elaborate on the rationale behind the two experiments and the inconsistent findings between them.
  • We thank the reviewer for the constructive comment. In the revised manuscript, we have elaborated the rationale behind two experiments (Lines 276-284) and discussed these inconsistent findings in detail (Lines 418-427).
  1. In discussion, the authors should further discuss the findings in combination with the adaptive control hypothesis.
  • We thank the reviewer for the valuable comment. In the revised manuscript, we have added a detailed discussion in combination with the adaptive control hypothesis (Line 383-388).

Once again, we thank the reviewer very much for spending the time reviewing our work.

Reviewer 3 Report

Comments and Suggestions for Authors

Linking Context to Language Switching: Effects of 2

Domain-General Conflicting Context on Comprehension-Based 3

Switch Costs 4

Lu Jiao a, Zejun Wang a, Xiaoting Duan b, Yingying Yu a and Cong Liu a

Introduction

Major comments:

The authors have minor grammar and flow issues. One major concern is that the introduction needs to provide more detail on more studies that semantic-conflicting contexts and switch costs. The authors are recommended to complete a more comprehensive search that would help support their aims.

In addition, the authors do not provide any detail about studies that examined domain-general conflicts. How is this defined, and how has domain-general context been examined/defined in prior studies? In addition, the authors are asked to describe similar studies that have explored the domain-context costs so that the reader can evaluate whether their exploration into this area is validated and necessary. There is not enough information provided to indicate any theoretical grounding to explain why we would need to explore domain-general contexts and their effect on switch costs. 

Throughout the manuscript, the phrasing of "noise context" should be revised as it's not consistent with other literature. Context most frequently refers to semantic information or linguistic information provided alongside the signal; background noise when presented as a masker is not considered "context" in any studies I am familiar with, so the authors should revise to say something like "...in the presence of background noise" or "when masked with noise" to refer to these conditions as done in experiment 1.

Similarly, "Masking" appears to be an important part of this study but it is not mentioned anywhere in the introduction.

1. l.26-28. This section appears to introduce the adaptive language control hypothesis, but additional detail and specificity would benefit this section.

2. l. 30-32. The Liu et al study seems important to the context of the study, so the authors may consider adding an example to explain hat "conflicting" and "nonconflicting" language contexts are, which are unclear as written.

3. l. 33. What is the specific context effect "revealed" by this study? Describing the methods that were used in the study to further validate this claim would help readers evaluate the work and its relevance to the present project.

4. The authors should consider adding "noise" somewhere to abstract and/or title since that seems to be a major focus of the purpose of the manuscript.

5. l. 52: "such as" should be revised to show that the following phrase is a detail of the prior clause, not an example of the prior clause.

6. l. 69-71. The authors' argument in this paragraph is that the switch costs are not consistent between a bilinguals' two languages. If that is correct, then Philipp & Hustegge citation does not provide support or example of this, as their work seems to focus on a different type of task. The claim that the switch costs are asymmetrical across languages is thus not supported as written. 

7. L.72-74. These studies provide some support for asymmetry, but the authors need to describe in additional detail. Specifically, was L1 > L2, or the opposite, or perhaps the cost effect was different for each language? What kind of asymmetry existed based on these studies?

8. L. 75. The authors cite "inconsistent results" but all the evidence provided only support one view; thus, no inconsistency is presented.

9. L87-89. The authors indicate that their goal is to "provide direct evidence that....". The authors are recommended to write in a way that shows they are neutral and unbiased by revising this phrasing to say something like: "...in the present research, our goal is to examine the effects of domain-general...".

10. L105: The authors should qualify "unbalanced" as this is a term that is controversial in current literature. 

2. Methods

Phrase "noise context" is confusing as the phrase 'context' refers typically to linguistic information not signal-based distraction. Perhaps "noise condition"?

I also have major concerns about the lack of specificity with the RT numbers, which appear to be whole numbers in the descriptive tables. The authors are advised to provide millisecond responses, which include tenths and hundredths columns so we can evaluate the fine-grained differences in RTs, which is customary in other RT research.

1. l.120 indicates that participants who performed <80% were removed from analysis. Why was this number chosen? Was it derived from previous studies or does it represent below average performance? My concern is that this seems subjective and like a random selection of the dataset.

2. l. 126, Table 1. Units for each of these items should be provided. Are the means for L2 AOA (8.23, for example), represented in years? It is unclear why Experiment 2 data is included here, or why it is italicized. The authors should remove Experiment 2 and reserve it for that section.

3. L.132. I am concerned with the fact that participants got to familiarize themselves with picture names. Would that have an impact on their abilities to complete the task above and beyond any effect of task switching?

4. L.137. Authors should specify if the 4 trial types are presented in random/counterbalanced order.

5. l. 149. Authors indicated they removed responses >2.5 SDs from the mean. The authors should clarify whether there was a lower boundary for removal, such as responses less than 100 or 200 ms in response which may indicate a spurious response. 

6. l.216. The authors again write that the task elicits "comprehension based" switch costs. I am unsure how the protocols of this study evoke any type of comprehension. Are the authors arguing that the semantic categorization task requires comprehension, such that slower comprehension is a result of greater switch costs in these conditions? I am concerned that the slower response to the task does not actually reflect slower comprehension; instead, it is entirely likely that there are other factors that make participants slower to respond to the task such as semantic interference effects or general language proficiency effects.

7. The rationale presented in l.242 for presenting weaker noise does not make sense. Why are the authors claiming their results are a result of such a signal to noise ratio? Why would changing the signal ratio make any difference and address their hypothesis? What are the "different amounts of conflicts" (since I understand there to only be 2 conditions in the study) and how would that be addressed by having weaker noise (there are still 2 conditions presumably)?

Experiment 2

1. l. 248. Authors should clarify if the participants in Experiment 2 were the same participants as those in experiment 1.

Table 2, 5, 

2. L. 284. It seems to me in Table 5 that the RTs are presented as whole seconds, without tenths of a seconds being indicated. Since the effect size is bound to very small, I have concerns that this is not sensitive enough to show an effect of reaction time. Additional ms latency is required to evaluate the work.

3. Lines 300-315. This exploratory analysis compares the two experiments, but I have major concerns about the integrity of this comparison given the fact that, among other issues, they are different participants of different n-sizes. This section is methodologically concerning to the point where its removal should be strongly considered.

4. L 326-330. The authors should add detail in this section. For example: "Comparing switch costs in various conflicting contexts with different amounts of noise" lacks any detail of what actual analyses were done with what actual conditions. Is this referring to the exploratory analysis?

General Discussion

1. l. 338: "across various contexts" - use of the word various here is vague and should be revised to specify which contexts specifically were different from which other specific contexts.

2. l.339. I am still unclear on what "comprehension based" differences mean.

Author Response

Response to Reviewer 3

Introduction

  1. The authors have minor grammar and flow issues. One major concern is that the introduction needs to provide more detail on more studies that semantic-conflicting contexts and switch costs. The authors are recommended to complete a more comprehensive search that would help support their aims.
  • We thank the reviewer for the important suggestion. In the revised manuscript, we have added more relevant studies on the effect of semantic-conflicting contexts (Lines 84-96).
  1. In addition, the authors do not provide any detail about studies that examined domain-general conflicts. How is this defined, and how has domain-general context been examined/defined in prior studies? In addition, the authors are asked to describe similar studies that have explored the domain-context costs so that the reader can evaluate whether their exploration into this area is validated and necessary. There is not enough information provided to indicate any theoretical grounding to explain why we would need to explore domain-general contexts and their effect on switch costs.
  • We thank the reviewer for the constructive comment. First, combined with previous studies, we have removed the phrasing of "domain-general conflicts", and clarified it explicitly as "background noise". Second, as suggested by the reviewer, we have added more empirical support and a theoretical basis on the background noise effect and its potential effect on switch costs (Lines 97-114).
  1. Throughout the manuscript, the phrasing of "noise context" should be revised as it's not consistent with other literature. Context most frequently refers to semantic information or linguistic information provided alongside the signal; background noise when presented as a masker is not considered "context" in any studies I am familiar with, so the authors should revise to say something like "...in the presence of background noise" or "when masked with noise" to refer to these conditions as done in experiment 1.
  • We thank the reviewer for the important suggestion. We have gone through the entire manuscript for editing and have changed the phrasing of "noise context" to "background noise".
  1. Similarly, "Masking" appears to be an important part of this study but it is not mentioned anywhere in the introduction.
  • Thanks for the valuable comment. In the revised manuscript, we have added the energetic masking phenomenon and introduced relevant studies to this (Lines 109-114).
  1. l.26-28. This section appears to introduce the adaptive language control hypothesis, but additional detail and specificity would benefit this section.
  • Thanks for the valuable comment. We have added details to this (Lines 26-31).
  1. l. 30-32. The Liu et al study seems important to the context of the study, so the authors may consider adding an example to explain hat "conflicting" and "nonconflicting" language contexts are, which are unclear as written.
  • Thanks for the comment. In the revised manuscript, we have elaborated on the study of Liu et al. (2019) and provided examples of "non-conflicting" and "conflicting" language contexts (Lines 35-38).
  1. l. 33. What is the specific context effect "revealed" by this study? Describing the methods that were used in the study to further validate this claim would help readers evaluate the work and its relevance to the present project.
  • We have elaborated the study of Liu et al. (2019) in the revised manuscript and provided the detailed findings of this study (Lines 32-40).
  1. The authors should consider adding "noise" somewhere to abstract and/or title since that seems to be a major focus of the purpose of the manuscript.
  • We appreciate the reviewer for the valuable comment. We have added "noise" to the title and abstract in the revised manuscript.
  1. l. 52: "such as" should be revised to show that the following phrase is a detail of the prior clause, not an example of the prior clause.
  • In the revised manuscript, we have removed the section to avoid misunderstanding.
  1. l. 69-71. The authors' argument in this paragraph is that the switch costs are not consistent between a bilinguals' two languages. If that is correct, then Philipp & Hustegge citation does not provide support or example of this, as their work seems to focus on a different type of task. The claim that the switch costs are asymmetrical across languages is thus not supported as written.
  • Thank you for this important clarification. In the revised manuscript, we have removed the Philipp & Hustegge citation.
  1. L.72-74. These studies provide some support for asymmetry, but the authors need to describe in additional detail. Specifically, was L1 > L2, or the opposite, or perhaps the cost effect was different for each language? What kind of asymmetry existed based on these studies?
  • In the revised manuscript, we have clarified the details of the asymmetrical pattern in the revised manuscript (e.g., Lines 93-94).
  1. L. 75. The authors cite "inconsistent results" but all the evidence provided only support one view; thus, no inconsistency is presented.
  • In the revised manuscript, we have removed this section and revised the Introduction section.
  1. L87-89. The authors indicate that their goal is to "provide direct evidence that....". The authors are recommended to write in a way that shows they are neutral and unbiased by revising this phrasing to say something like: "...in the present research, our goal is to examine the effects of domain-general...".
  • As suggested by the reviewer, we have revised this sentence (Line 129).
  1. L105: The authors should qualify "unbalanced" as this is a term that is controversial in current literature.
  • Thanks for this comment. In the revised manuscript, we have clarified the "unbalanced bilinguals with dominant Chinese" (Line 133), and reported the detailed information in the Methods section (Line 152-157 in Experiment 1, Lines 290-296 in Experiment 2).

Methods

  1. Phrase "noise context" is confusing as the phrase 'context' refers typically to linguistic information not signal-based distraction. Perhaps "noise condition"?
  • Thanks for this important clarification. In the revised manuscript, we have changed the phrasing of "noise context" to "noise condition".
  1. I also have major concerns about the lack of specificity with the RT numbers, which appear to be whole numbers in the descriptive tables. The authors are advised to provide millisecond responses, which include tenths and hundredths columns so we can evaluate the fine-grained differences in RTs, which is customary in other RT research.
  • Thanks for this suggestion. In the revised manuscript, we have provided millisecond responses including tenths and hundredths columns in the descriptive tables (see Table 2 and 6).
  1. l.120 indicates that participants who performed <80% were removed from analysis. Why was this number chosen? Was it derived from previous studies or does it represent below average performance? My concern is that this seems subjective and like a random selection of the dataset.
  • Thanks for the valuable comment. The criterion of data rejection was in line with previous studies (e.g., Liu, Timmer, Jiao, & Wang, 2019).
  1. l. 126, Table 1. Units for each of these items should be provided. Are the means for L2 AOA (8.23, for example), represented in years? It is unclear why Experiment 2 data is included here, or why it is italicized. The authors should remove Experiment 2 and reserve it for that section.
  • We have added units for each of these items in Table 1 (Line 157), and added a separate table (Table 5, Line 296) for presenting participant's information of Experiment 2.
  1. L.132. I am concerned with the fact that participants got to familiarize themselves with picture names. Would that have an impact on their abilities to complete the task above and beyond any effect of task switching?
  • We thank the reviewer for this valuable comment. This is a common case in language switching research (e.g., de Bruin, Roelofs, Dijkstra, & FitzPatrick, 2014; Jiang, Meng, & Chen, 2024; Liu & Chaouch-Orozco, 2023; Tong, Kong, Wang, Liu, Li, & He, 2019), thus, to diminish the number of errors in the bilingual switching experiment.
  1. L.137. Authors should specify if the 4 trial types are presented in random/counterbalanced order.
  • Thanks for this suggestion. For quiet and noise conditions, all trials were pseudorandomized with no more than four trials of the same type (switch or repeat), language (L1 or L2), or animacy category in a row (Lines 182-184).
  1. l. 149. Authors indicated they removed responses >2.5 SDs from the mean. The authors should clarify whether there was a lower boundary for removal, such as responses less than 100 or 200 ms in response which may indicate a spurious response. 
  • We have also removed the responses faster than 200 ms, and added this in the revised manuscript (Line 224).
  1. l.216. The authors again write that the task elicits "comprehension based" switch costs. I am unsure how the protocols of this study evoke any type of comprehension. Are the authors arguing that the semantic categorization task requires comprehension, such that slower comprehension is a result of greater switch costs in these conditions? I am concerned that the slower response to the task does not actually reflect slower comprehension; instead, it is entirely likely that there are other factors that make participants slower to respond to the task such as semantic interference effects or general language proficiency effects.
  • We thank the reviewer for this comment. The phrasing of "comprehension-based/production-based switching" is common in the bilingualism literature (e.g., Liu, Timmer, Jiao, & Wang, 2020; Coumel et al., 2024). In order to avoid suggestions, we have changed the phrasing of "comprehension-based language switching" to "language switching in comprehension".
  1. The rationale presented in l.242 for presenting weaker noise does not make sense. Why are the authors claiming their results are a result of such a signal to noise ratio? Why would changing the signal ratio make any difference and address their hypothesis? What are the "different amounts of conflicts" (since I understand there to only be 2 conditions in the study) and how would that be addressed by having weaker noise (there are still 2 conditions presumably)?
  • We thank the reviewer for the valuable comment. In the revised manuscript, we have removed the phrasing of "different amounts of conflicts" to avoid misunderstanding, and elaborated the rationale behind two experiments (Lines 276-284)

Experiment 2

  1. l. 248. Authors should clarify if the participants in Experiment 2 were the same participants as those in experiment 1.
  • We have clarified that the participants in Experiment 2 were another group of bilinguals, not the same group participating in Experiment 1 (Line 287).
  1. L. 284. It seems to me in Table 5 that the RTs are presented as whole seconds, without tenths of a seconds being indicated. Since the effect size is bound to very small, I have concerns that this is not sensitive enough to show an effect of reaction time. Additional ms latency is required to evaluate the work.
  • As suggested by the reviewer, we have provided millisecond responses including tenths and hundredths columns in the descriptive tables (see Table 2 and 6).
  1. Lines 300-315. This exploratory analysis compares the two experiments, but I have major concerns about the integrity of this comparison given the fact that, among other issues, they are different participants of different n-sizes. This section is methodologically concerning to the point where its removal should be strongly considered.
  • As suggested by the reviewer, we have moved this section in the revised manuscript.
  1. L 326-330. The authors should add detail in this section. For example: "Comparing switch costs in various conflicting contexts with different amounts of noise" lacks any detail of what actual analyses were done with what actual conditions. Is this referring to the exploratory analysis?
  • Yes, this sentence is referring to the exploratory analysis. As suggested by the reviewer that the "exploratory analysis" section should be removed (see comment 26), we have also removed these related discussions.

General Discussion

  1. l. 338: "across various contexts" - use of the word various here is vague and should be revised to specify which contexts specifically were different from which other specific contexts.
  • We have revised this sentence (Line 358).
  1. l.339. I am still unclear on what "comprehension based" differences mean.
  • As we reply to the above comment (comment 22), aiming to avoid misunderstanding, we have changed the phrasing of "comprehension-based switch costs" to "switch costs in comprehension".

Once again, we thank the reviewer very much for spending the time reviewing our work.

Round 2

Reviewer 1 Report

Comments and Suggestions for Authors

Please read the file, thanks.

Author Response

After the first round of revisions this paper was supplemented with detailed additions to both the introduction section as well as the experiments and discussion parts, which improved the article as a whole. And the article was logically smoother and better organized, with appropriate detail and a complete account of the key experiments. This study is helpful for expanding the adaptive control hypothesis of language switching based on language understanding in different sound environments. In spite of this, there are still some small problems that need to be carefully revised by the authors.

Experiment 1

  1. The use of word datasets for experimental materials should be properly labeled with references to cited papers or databases.
  • Thanks for the comment. We have added the reference to the cited database in the revised manuscript (Line 163).
  1. The discussion of Experiment 1 mentions that the switch costs with the presence of white noise were smaller than those in a quiet condition can be explained by the stochastic resonance phenomenon and we suggest that there could have a brief addition to the stochastic resonance phenomenon, that could be better.
  • We thank the reviewer for the valuable suggestion. In the revised manuscript, we have provided a brief addition to the stochastic resonance phenomenon in the Discussion of Experiment 1 (Lines 273-277).

Discussion

  1. The end of the Abstract section mentioned that the study extended the adaptive control hypothesis, but the General Discussion section does not provide a detailed description of how the study extended and which part of the adaptive control hypothesis be extended.
  • We thank the reviewer for the valuable comment. In the revised manuscript, we have elaborated on how our findings extend the adaptive control hypothesis (Line 387-391).
  1. The discussion section can add an outlook on the shortcomings of this study and future research directions, that could make the discussion more completed.
  • Thanks for the valuable suggestion. We have discussed our shortcomings and future research directions in the revised manuscript (Lines 436-442).

Once again, we thank the reviewer very much for spending the time reviewing our work.

Reviewer 2 Report

Comments and Suggestions for Authors

It is a great study. No question any more.

Author Response

Comment: It is a great study. No question any more.

Response: We thank the reviewer again for the valuable comments.

Reviewer 3 Report

Comments and Suggestions for Authors

No comments. Authors have adequately addressed concerns.

Author Response

Comment: No comments. Authors have adequately addressed concerns.

Response: We thank the reviewer again for their valuable comments.